# Clinical Evaluation of Sleep Disorders in Parkinson’s Disease

**DOI:** 10.3390/brainsci13040609

**Published:** 2023-04-03

**Authors:** Fulvio Lauretani, Crescenzo Testa, Marco Salvi, Irene Zucchini, Francesco Giallauria, Marcello Maggio

**Affiliations:** 1Department of Medicine and Surgery, University of Parma, 43126 Parma, Italy; 2Clinic Geriatric Unit and Cognitive and Motor Center, Medicine and Geriatric-Rehabilitation Department, University-Hospital of Parma, 43126 Parma, Italy; 3Department of Translational Medical Sciences, “Federico II” University of Naples, Via S. Pansini 5, 80131 Naples, Italy

**Keywords:** Parkinson’s disease, sleep disorders, Braak’s stages, treatment, drugs

## Abstract

The paradigm of the framing of Parkinson’s disease (PD) has undergone significant revision in recent years, making this neurodegenerative disease a multi-behavioral disorder rather than a purely motor disease. PD affects not only the “classic” substantia nigra at the subthalamic nuclei level but also the nerve nuclei, which are responsible for sleep regulation. Sleep disturbances are the clinical manifestations of Parkinson’s disease that most negatively affect the quality of life of patients and their caregivers. First-choice treatments for Parkinson’s disease determine amazing effects on improving motor functions. However, it is still little known whether they can affect the quantity and quality of sleep in these patients. In this perspective article, we will analyze the treatments available for this specific clinical setting, hypothesizing a therapeutic approach in relation to neurodegenerative disease state.

## 1. Introduction

Parkinson’s disease affects 1–2% of the population over 60 years of age. Its incidence and prevalence are constantly increasing, especially in the last decades of life [1,2]. Due to the age of onset and the complexity and manifestation of symptoms, a comprehensive geriatric assessment is considered the best-suited approach for this disease. 

Recently, Parkinson’s disease has been recognized as a multisystemic disorder, and while the inflammatory pathogenesis for years was only the prerogative of the cardiovascular system [3], more and more evidence indicates that even neurodegeneration is secondary to genetic causes and to alterations of the inflammatory state [4,5]. 

This is an important upgrade in comparison with preexisting theories focusing only on the dopaminergic neurons of the substantia nigra [6,7,8,9,10,11,12,13]. Thus, in the absence of treatments attenuating or reverting neurodegeneration, the symptomatic management of the disease, targeting both motor and nonmotor symptoms, has become very important. Over the years, the therapeutic approach to motor symptoms has produced surprising results in patients who fully respond to dopamine replacement therapy [14].

However, more remains to be done for non-motor symptoms. Given the growing number of elderly and multimorbid patients and the increasingly demanding management of chronic and advanced stages of the disease, this approach is particularly relevant to the quality of life of patients and their caregivers [15].

Therefore, regarding the plethora of drugs available for the treatment of Parkinson’s disease, the choice must be accurate and suited to the patient’s needs. Any treatment worsening the quality of life of patients or their caregivers should be avoided [16].

Under the umbrella of non-motor symptoms, sleep disturbances are among the most common and those with the greatest impact on a patient’s quality of life [17,18]. It is estimated that nearly half of the patients with Parkinson’s disease suffer from sleep disturbances. Surprisingly, many patients underreport the symptom simply because they do not consider it as part of the disease [17,18,19,20]. Thus, the correct classification of sleep disorders in Parkinson’s disease is relevant. The aim of this perspective article is to provide a correct framing of sleep disturbances in Parkinson’s disease in relation to the Braak’s scale. Finally, we will try to identify the most correct treatment in relation to the disease state.

## 2. Sleep Disorders in Parkinson’s Disease: A Motley Melting Pot

Symptom-wise, sleep disturbances in Parkinson’s occur in variegate ways. In this section, we will analyze the different ways in which sleep disorders can occur. Insomnia is the most frequent sleep disorder in Parkinson’s disease with the prevalence, ranging from 30% to 80% of affected patients, increasing as the disease progresses [21,22]. Insomnia is difficulty in initiating or maintaining sleep. In patients with Parkinson’s disease, difficulty in maintaining sleep (with early awakenings and sleep fragmentation) is more frequently described than difficulty in initiating sleep [23]. The sleep and circadian rhythm regulatory centers are affected by the neurodegeneration typical of Parkinson’s disease, which lays the pathophysiological basis for the development of insomnia [24]. This substrate combined with the presence of the off symptoms contributes to the development and aggravation of insomnia as the disease progresses [25,26].

As for the diagnosis, in addition to an accurate medical history, clinicians have at their disposal a series of questionnaires, some of these specifically validated for Parkinson’s disease (PDSS, PDSS-II, and SCOPA). In the most severe cases or the differential diagnosis of comorbidities, polysomnography is indicated [27,28]. Another way of presentation of sleep disturbances in Parkinson’s disease is restless legs syndrome (RLS). A meta-analysis clearly shows that this syndrome affects about 15% of Parkinsonian patients [29]. This disorder occurs with the urge to move the legs and is usually associated with leg discomfort. Symptoms generally begin in the late afternoon or during the night, causing a great deal of discomfort to the patient and her/his partner. Regarding the etiology, there are three pathogenetic hypotheses: (one) in relation to the response to dopaminergic supplementation, Parkinson’s disease and RLS share a common dopaminergic degeneration and a possible genetic connection [30]; (two) RLS in Parkinson’s disease has a different etiology than idiopathic RLS; (three) RLS and Parkinson’s disease are two different pathologies [31]. 

As evident from these hypotheses, there is also a type of RLS that occurs independently of Parkinson’s disease [31]. The diagnostic criteria of RLS are described in the International Classification of Sleep Disorders [32]. Particular attention is needed in the diagnosis of this syndrome since it is capable of imitating other common symptoms, especially in elderly patients such as myalgia, leg cramps, and arthritis. Another sleep disorder typical of Parkinson’s disease is rapid eye movement sleep behavior disorder (RBD). This disorder is parasomnia and consists of repeated vocalizations during sleep or complex motor behaviors during REM sleep. Polysomnographic studies have shown that the loss of muscle tone typical of the REM phase is lost in this disorder [32]. Approximately 24% of patients with Parkinson’s disease are affected by this disorder, compared with 3.4% of affected individuals in the general population [33]. Similar rates were also found in another study [34]. It is important to highlight how idiopathic RBD is considered a strong predictor of synucleinopathies. A multicenter study reported a conversion rate from RBD to Parkinson’s disease of approximately 6.3% annually and 73.5% after a 12-year follow-up period [35]. RBD precedes the onset of Parkinson’s disease by about 13 years [36]. As far as pathophysiology is concerned, RBD has been associated with dysfunction in the pontomedullary and other structures regulating REM sleep, in particular, the locus coeruleus [37]. Also, in this case, in addition to the diagnostic criteria of the *International Classification of Sleep Disorders*, a specific diagnostic questionnaire was drawn up [38]. 

A consequence of sleep disturbances in patients with Parkinson’s disease is excessive daytime sleepiness (EDS) which occurs in 20 to 75% of patients with Parkinson’s disease [39,40,41]. This disorder consists of difficulty staying awake and alert during the day [32]. An accredited etiopathogenetic hypothesis ascribes this disorder to hypothalamic neurodegeneration and different nuclei of the brain stem responsible for the sleep–wake cycle [42]. As regards the diagnostic process, the Epworth sleepiness scale (ESS) is generally used as a screening tool. It is important to exclude other diseases that can cause daytime sleepiness such as RLS, OSA, and RBD. Finally, sleep disorders related to respiratory problems, in particular obstructive sleep apnea, should be accounted giving the prevalence of 20–60% of patients with Parkinson’s disease [43,44]. In Parkinsonian patients, laryngopharyngeal motor dysfunction with occlusion of the upper respiratory tract is the cause of obstructive apneas [45]. As far as diagnosis is concerned, polysomnography is the gold standard exam, also validated in patients with Parkinson’s disease [46,47]. Sleep disorders in Parkinson’s disease are a heterogeneous melting pot of disorders. It is difficult to draw a guideline to guide clinicians in the treatment of these pathologies since many patients do not even ascribe the problem to Parkinson’s disease. Except for RBD, which can be framed as a prodrome of Parkinson’s disease, the rest of the sleep disorders generally present with a more advanced state of disease. It will be the task of the clinician who, with a careful history and a comprehensive assessment, will be able to diagnose and treat these disorders. An upgrade toward a comprehensive assessment of Parkinson’s disease patients cannot be postponed. There is increasing evidence that sleep disturbances not only correlate with a worse quality of life but also trigger a pathophysiological mechanism that exacerbates major depressive states [20]. Especially in the later stages of life with Parkinson’s disease, depression and nonmotor symptoms, rather than motor symptoms, have a greater impact on the quality of life of patients [48]. Therefore, patients with Parkinson’s require a comprehensive assessment to stop this vicious circle (neurodegenerative disease -> depression -> neurodegenerative disease) that, in the long term, leads to disability [49]. It will be interesting in the future to try to identify the primum movens of this vicious circle, also in consideration of its pathogenetic affinity with sleep disorders.

## 3. The Braak Scale: An Old Staging with a New Awareness 

For more than 20 years, Braak and colleagues [42] have postulated the hypothesis of progressive neurodegeneration in the etiology of sporadic Parkinson’s disease, and although there are numerous scales for staging Parkinson’s [50], the Braak scale is the one that best explains the pathophysiology of the disease. Regardless of the underlying etiological cause, over time, this hypothesis has been examined in various clinical and preclinical settings and was recently confirmed [51]. The concept of progressive neurodegeneration that inexorably advances and affects more and more brain areas is supported by the clinical manifestations of the disease. Six microscopically additive disease stages are described, with typical histological lesions (Lewy neurites and Lewy bodies): (one) lesions in the dorsal IX/X motor nucleus and/or intermediate reticular zone; (two) lesions in caudal raphe nuclei, gigantocellular reticular nucleus, and coeruleus—subcoeruleus complex; (three) midbrain lesions, particularly in the pars compacta of the substantia nigra; (four) prosencephalic lesions. Cortical involvement is confined to the temporal mesocortex (transentorhinal region) and allocortex (CA2-plexus). The neocortex is unaffected; (five) lesions in high-order sensory association areas of the neocortex and prefrontal neocortex; (six) lesions in first-order sensory association areas of the neocortex and premotor areas, occasionally mild changes in primary sensory areas and the primary motor field. 

It should be emphasized that the motor symptoms appear during the late phase of the disease progression, Braak stages 3–4 (39). The long prodromal phase corresponds to a neurodegeneration that remains in the subclinical state. RBD, which by many authors is considered a prodrome of Parkinson’s disease, is caused precisely by a degeneration at the level of the locus coeruleus which is affected in the initial stages of the disease (Braak stage two). Unlike RBD, other sleep disorders generally occur at or shortly after the onset of motor symptoms. The mode of presentation varies from patient to patient but with the progression of the disease and the worsening of motor symptoms less controlled by pharmacological treatment, there is a worsening presentation of sleep disturbances. This, almost schematic, trend must guide the clinical approach to stabilizing the sleep–wake cycle in the prodromal phases of the disease, to prevent dopaminergic neurodegeneration in the intermediate phases of the disease up to treatment with drugs that also target cognitive impairment associated-conditions during the final stages of the disease.

## 4. Clinical Implications and Available Treatments for Sleep Disorders in Parkinson’s Disease 

The main clinical implication of sleep disorders is the major negative impact on the quality of life of patients with Parkinson’s disease. Reduced quality of life very often results in a greater tendency to develop a mood disorder. In recent years, clinicians have begun to focus their attention on treating non-motor symptoms, particularly depression. Recent evidence indicates that drugs such as SSRI and SNRI, in young subjects’ tricyclic antidepressants, dopamine agonists, and behavioral therapy have good efficacy in the treatment of depression in patients with Parkinson’s disease. 

Insomnia and depression are closely related to Parkinson’s disease [23]. Patients with insomnia usually have a more advanced state of disease and are more prone to symptoms due to the wearing-off of the levodopa effect. They also show problems such as autonomic dysfunction, hallucinations, and postural instability [23,52]. The correct treatment of insomnia in Parkinson’s disease starts with the careful collection of patients’ histories. Insomnia can occur during the night as an end-of-dose effect of dopamine. For this reason, the use of an additional tablet of levodopa, prolonged-release levodopa, or a dopamine agonist finds more and more space in this clinical setting. Recent evidence indicates that drugs such as eszopiclone and melatonin also find their place in the treatment of insomnia related to Parkinson’s disease, especially in the early phase of the disease [53]. 

Related to insomnia or the underlying cause of insomnia, is restless leg syndrome (RLS). Dopamine agonists such as pramipexole, rotigotine, and ropinirole have their therapeutic rationale for treating it [54,55,56]. Careful attention is needed in the use of these drugs since, in some cases, they lead to a worsening of nocturnal symptoms after an initial benefit or could produce impulse compulsive disorders (ICDs) with nocturnal activity and sleep interruption [57]. In these cases, the treatment must be suspended and replaced either with a long-acting dopaminergic drug or with a drug acting beyond dopamine stimulation [58]. In this regard, good evidence of efficacy has been found with the use of gabapentin or pregabalin [59,60].

The clinical implications of rapid eye movement sleep behavior disorder (RBD) are very important because when this syndrome occurs as a comorbidity in Parkinson’s disease, it is associated with increased motor dysfunction, hallucinations, cognitive impairment, and autonomic dysfunction, especially in the advanced phase of the disease [61,62]. There is still little evidence of an effective treatment for RBD, though some evidence of efficacy has been obtained with melatonin or clonazepam [63,64]. 

Older age is characterized by a worse presentation of disease symptoms and is also associated with excessive daytime sleepiness (EDS). EDS is often also the consequence of the breathing disorders associated with Parkinson’s disease. Once again, a correct diagnosis is important since in OSAs independent of Parkinson’s disease, the treatment of choice is C-PAP, while in OSAs caused by Parkinson’s disease, the use of controlled release formulations of carbidopa/levodopa at bedtime may improve symptoms [65].

It is evident that there is a lack of effective treatments in most of the analyzed conditions. Furthermore, to set up a correct therapeutic procedure, a precise diagnosis is required, together with a good knowledge of the different ways in which sleep disorders can occur in relation to the degree of neurodegeneration. Finally, it is necessary to modulate the therapeutic treatments based on the caregiver’s feedback, especially in the most advanced stages of the disease. The application of these procedures allows the avoidance of pharmacological overshooting that can lead to a decrease in alertness with very serious consequences such as inhalations of ingests and consequent pneumonia.

## 5. Available Drugs and State of Art of Treatment 

Although sleep disorders in Parkinson’s disease have been a clinical problem receiving interest from the scientific community for many years, no effective and lasting therapeutic approach has still been validated. Probably, the primary reason for the existing gap between clinical need and lack of adequate therapies is due to the heterogeneity, not only of the clinical manifestations but also of the type of patient in which these clinical manifestations appear. With the need to improve patients’ symptoms, numerous therapeutic approaches have been attempted, whose clinical efficacy is substantially anecdotal and not evidence-based. The complexity of the patients and the heterogeneous responses to the different pharmacological treatments must direct clinicians to have an excellent knowledge of the numerous drugs available to avoid exposing the patient to the sedative actions of these drugs. In this clinical setting, there is no “one drug fits all”, though only an adequate comprehensive assessment oriented to the patient and the living environment can help physicians towards a multidomain treatment, including prescribing or sometimes deprescribing certain medications.

Among the first drugs used in this clinical context are melatonin and its synthetic derivative (Ramelteon and Agomelatine), agonists of melatonin receptors. Melatonin is an agonist of the MT1, MT2, and MT3 receptors, has a half-life of about 4 h, and is one of the first drugs used in patients with insomnia. Experiences in patients with Parkinson’s disease indicate that melatonin can improve the quality of sleep [66,67]. We have previously found that in patients with Parkinson’s disease, insomnia does not manifest itself as difficulty falling asleep, but rather as difficulty staying asleep. In this regard, various therapeutic approaches are available, and one of these is the use of prolonged-release melatonin. In two recent studies, a 2 mg dose of prolonged-release melatonin was associated with significant improvements in night-time frequency and nocturnal voided volumes, and beneficial effects on sleep quality with improved nonmotor symptoms and quality of life in PD patients [68,69]. In patients in whom a coexistence between sleep disorder and depression of mood emerges at the visit, the use of agomelatine at a dosage of 12.5 mg, titrated up to 50 mg, could be useful [70]. Ramelteon is the synthetic derivative of melatonin mostly used in patients with Parkinson’s disease. It acts as an MT1 and MT2 receptor agonist and has a half-life of approximately 2.5 h. At an 8 mg dose, ramelteon was effective in the treatment of sleep disorders in Parkinson’s patients, particularly in RBD [71].

There is little data in the literature on the use of benzodiazepines in patients with Parkinson’s disease, although in clinical practice they are often used above all for the relief of depressive symptoms and for their hypnotic action. These drugs act as positive allosteric modulators of the GABA receptor and differ substantially in the length of half-life. In light of the scarcity of significant evidence, the use of benzodiazepines in patients with Parkinson’s disease should be weighed on a case-by-case basis, especially in relation to their side effects, one of which is inhalation pneumonia. A case-control study of over 550,000 patients found that benzodiazepine use is associated with an increased risk of pneumonia in elderly patients with Parkinson’s disease [72]. It is therefore essential that the indication for the use of these drugs, in this particular setting, should be managed by expert physicians. Among the various benzodiazepines, more consistent data have emerged on the use of clonazepam for RBD. Clonazepam is a benzodiazepine with a long half-life (30–40 h) and is indicated by many as the first-line treatment in RBD. The evidence for the use of clonazepam in RBD is supported by studies with small sample sizes, some of which did not reach statistical significance when compared to a placebo [73,74,75]. Considering the long half-life and the possibility of accumulation phenomena, especially in elderly patients, the use of clonazepam in this clinical setting requires careful attention. Among the antidepressants, the one with the greatest sedative action, trazodone, is often used off label as a hypnotic inducer in elderly patients. This molecule acts as an antagonist of the serotonin 5HT2a/c receptor, the stimulation of which has a known antidopaminergic effect. In relation to this function, trazodone improved depressive symptoms and motor function in patients with Parkinson’s disease [76]. In patients with Parkinson’s, there are few data concerning the use of trazodone as a hypnotic inducer. However, a very recent experience conducted on 31 patients demonstrated its efficacy and tolerability at a sedative hypnotic dosage (50 mg) in this clinical setting [77]. The efficacy of trazodone as a hypnotic inducer is probably also due to its biphasic half-life with a first phase of 3–6 h and a second phase of 5–6 h. This biphasic effect is particularly welcome in Parkinsonian patients where insomnia is mainly due to difficulty staying asleep.

Among the nonbenzodiazepine allosteric modulators of the GABA receptor, z-compounds are often used in clinical practice as hypnotic agents due to their reduced sedative effects and therapeutic handling [53,78]. Randomized controlled trials on the use of these drugs in the setting of our interest are scarce. More significant experiences have been made with eszoplicone, which has demonstrated excellent tolerance and good clinical efficacy as a hypnoinducer in Parkinson’s patients [79]. 

Among the nonbenzodiazepine allosteric modulators of the GABA receptor, drugs such as gabapentin and pregabalin have a more defined and codified therapeutic niche for sleep disorders in Parkinson’s patients. In fact, there are numerous pieces of evidence, especially for the long-release pharmaceutical formulation of gabapentin, of their effectiveness in restless leg syndrome. In geriatric patients, these drugs are well-tolerated, though they require careful evaluation of renal function before starting and during treatment [80,81,82].

Few randomized controlled trials are available for the use of antipsychotic drugs in patients with sleep disorders and Parkinson’s disease. The complexity of using these drugs is mainly due to the side effects, in particular sedation, and the need for clinical monitoring at the time of cardiac repolarization, exposing the patient to a greater risk of arrhythmias. There is no indication of these drugs in the initial stages of the disease, while their use will be more appropriate in the very advanced stage. Quetiapine is an atypical antipsychotic drug with low receptor specificity. Antagonized receptors include the histamine H1 receptor and serotonin 2A receptor. Consequently, sedation is intrinsic to the drug’s activity. Due to this receptor specificity, attempts have been made to use quetiapine for the treatment of insomnia, regardless of the presence of Parkinson’s disease, though the results due to the sedative effects, often in the presence of the other approved drugs for insomnia, are largely disappointing and the benefits of using quetiapine do not outweigh the risks [83]. The results of an open-label study have demonstrated that quetiapine can find its place in the treatment of insomnia in patients with Parkinson’s [84]. However, these results need to be confirmed with an appropriate study design that includes the comparison with placebo control or with drugs already approved for insomnia and using an adequate sample size. The effects of quetiapine in improving visual hallucinations in Parkinson’s patients are not related to a normalization of sleep architecture [85]. The effects of clozapine on sleep have not been specifically studied in patients with Parkinson’s disease, though its use may consolidate sleep in psychiatric patients [86].

In patients with Parkinson’s dementia, and consequent behavioral disturbances, low-dose clozapine may have a clinical indication, especially in patients where behavioral disturbances are particularly accentuated [87]. The use of other antipsychotic drugs in patients with sleep disorders and Parkinson’s disease has very little evidence-based validation in the literature and, consequently, their use must be weighed on a case-by-case basis. The use of antipsychotics in Parkinson’s disease is especially indicated in the treatment of psychosis in patients who have a very advanced state of the disease. However, compliance with therapy is low, and about one-third of patients prematurely terminate therapy due to both the side and antidopaminergic effects [88]. 

Pimavanserin is the most recently developed antipsychotic and has a peculiar mechanism of action that makes it substantially inactive on dopamine receptors. It acts as an antagonist and inverse agonist of serotonin 2A and 2C receptors. It finds indication above- all for hallucinations and delusions associated with psychoses related to Parkinson’s dementia, a fact now corroborated in the literature [89]. Already in phase 1/2 studies, some evidence indicated that pimavanserin could have an effect objectively assessed on the sleep rhythm [90]. Recent findings indicate that this new treatment may improve the quality of sleep both in patients treated for major depressive disorder and in those with psychosis in Parkinson’s disease [91,92].

Among the tricyclic antidepressants, doxepin has been shown to be effective in improving the quality of sleep in patients with Parkinson’s disease, though its use in this setting is not widespread [93].

Despite its efficacy in the treatment of primary insomnia and the prevention of delirium in hospitalized patients [94], Suvorexant has not yet found a validated clinical indication in patients with sleep disorders and Parkinson’s disease. Numerous scientific pieces of evidence justify the use of antidepressant drugs for Parkinson’s disease. However, it is still difficult in this setting to identify a treatment that has independent effects on sleep disturbances alone, rather than depression, given the close relationship between these two conditions in Parkinson’s disease. In addition to the drugs already highlighted, venlafaxine also seems to have a role in this clinical setting [95,96].

An off phase during the night can manifest itself as insomnia and modulation of dopaminergic therapy can be the best treatment. It is therefore essential to frame the sleep disorder presented by the patient with a correct medical history also detailed in the history of pharmacological therapy. There are numerous studies in the literature that have tried to endorse this therapeutic attitude. The use of a dose of levodopa upon awakening during the night as the main therapeutic action in insomnia linked to Parkinson’s disease is an approach that has yet to be validated in the literature, though it is certainly supported by numerous indirect evidence, which indicates that more constant dopaminergic stimulation is effective in this regard. Treatment with a levodopa-carbidopa gastrointestinal gel that achieves a constant therapeutic drug plasma concentration was shown in one study to improve sleep disturbances together with other symptoms in Parkinson’s disease [97]. 

Dopaminergic agonists are known to have a longer half-life and are less subject to change in pharmacokinetics than levodopa. A transdermal system for the release of rotigotine in patients complaining of sleep disturbances has shown how this treatment could improve the quality of sleep by reducing nocturnal awakenings and improving motor performance upon morning awakening [98]. In patients with advanced Parkinson’s disease, both the immediate-release and prolonged-release pharmaceutical formulations of pramipexole have been shown to be effective in improving the subjective quality of sleep [99]. Ropinirole as an add on to levodopa therapy has also been shown to improve subjective symptoms in patients with Parkinson’s disease, both with immediate-release pharmaceutical formulations and with prolonged-release pharmaceutical formulations in different disease stages [100,101]. Cabergoline therapy as an add on to levodopa monotherapy has also been shown to be effective in improving both polysomnographic parameters and the subjective quality of sleep in patients with idiopathic Parkinson’s disease [102]. 

Dopaminergic stimulation has also been shown to be effective in treating sleep disorders in Parkinson’s disease other than insomnia. In a randomized controlled study of over 300 patients, the efficacy of levodopa and cabergoline in the treatment of RLS was compared; the study showed a greater efficacy on symptoms for the cabergoline treatment, while patients in the levodopa treatment group reported better tolerability [103]. Pramipexole, rotigotine, and ropinirole have also shown good efficacy in controlling RLS symptoms [104,105,106]. Finally, both immediate-release and extended-release ropinirole have been shown to have a significant effect in mitigating daytime sleepiness episodes in EDS [107].

To support our hypothesis postulating the undertreatment of sleep disorder in Parkinson’s disease, two authors (FL and CT) separately screened major medical databases in search of clinical randomized controlled trials conducted in the setting of our interest. The keywords “Parkinson’s disease” and “sleep disorder” and all possible combinations were used to screen the Medline, EMBASE, and Scopus databases, and 5786 articles were screened. The article selection process is summarized in Figure 1 according to a PRISMA diagram. In Table 1 are summarized all the randomized controlled trial regard the setting of our interest. 

## 6. Right Medication at Right Disease State

Figure 2 is the mainstay of our therapeutic proposal. It is essential to associate the correct treatment for sleep disturbance with a certain disease state. In the prodromal phases, there are medications such as melatonin or antidepressants such as trazodone or mirtazapine. In the symptomatic phases for motor disturbances, treatment with additional doses of levodopa in the night or with prolonged release formulations of levodopa or dopamine agonists may be useful, supporting the idea that the sleep disorder could be a “non-motor off state”. Finally, in the final stages of the disease, in which modest cognitive impairment with behavior disorders can be present, the use of antipsychotics finds space. In this advanced phase of the disease, normally, many drugs affecting cognition, depression, anxiety, behavior symptoms, and mobility are prescribed with a tailored therapy that could be specific for each patient. This topic was recently emphasized in the context of psychosis, where authors underlined different sleep disorders throughout the course of the disease and different psychosis stages showed distinct abnormalities in sleep quality, architecture, and spindles [128]. These findings altogether suggest that sleep disorders could become a core treatment in different neurodegenerative diseases, such as psychosis, Parkinson’s disease, and dementia [129,130]. In this clinical context, a correct pharmacological treatment can only take place after the comprehensive evaluation of the patient accompanied by a correct pharmacological history. Especially in elderly patient, drug treatment can itself be a cause of clinical worsening and hospitalizations that negatively impact the patient’s prognosis.

In the advanced stage of neurodegenerative diseases, sleep disorders probably represent a challenge for physicians [131,132], especially geriatricians, where the balance between deprescribing or drug appropriateness could become the key element for maintaining a patient at home.

## 7. Conclusions 

In the world of the geriatric population, polypharmacotherapy frequently occurs. Epidemiological trends indicate that more and more elderly patients are exposed to the risk of being overtreated without a real clinical benefit and to a greater risk of adverse clinical consequences. We have already described how the improper use of sedative drugs in Parkinson’s disease such as benzodiazepines can expose patients to a greater risk of inhalation pneumonia, as reported for other drug classes such as antipsychotics and antidepressants. The treatment of sleep disorders in Parkinson’s disease cannot benefit from dichotomous indications. This clinical problem is strictly dependent on factors such as the stage of the disease and the patient’s insight into the problem. The scenario totally changes in the advanced stages of Parkinson’s dementia. In light of the considerations made, and the available evidence, it is possible to make an indication of a therapeutic attitude rather than a therapeutic indication. The sleep disorders in a patient with Parkinson’s disease must be viewed from a multidimensional perspective. It is essential to indicate therapeutic treatments that are biologically consistent with the stage of the disease. Especially in an elderly patient, the therapeutic indications must be balanced with other pharmacological treatments and the patient’s comorbidities, avoiding the exposure of the patient to sedation and other relevant harmful side effects. In this setting, it will be important in the future to design randomized controlled trials that take into account the heterogeneity of the elderly population with Parkinson’s disease and the different types and modalities of presentation of sleep disorders.

## Figures and Tables

**Figure 1 brainsci-13-00609-f001:**
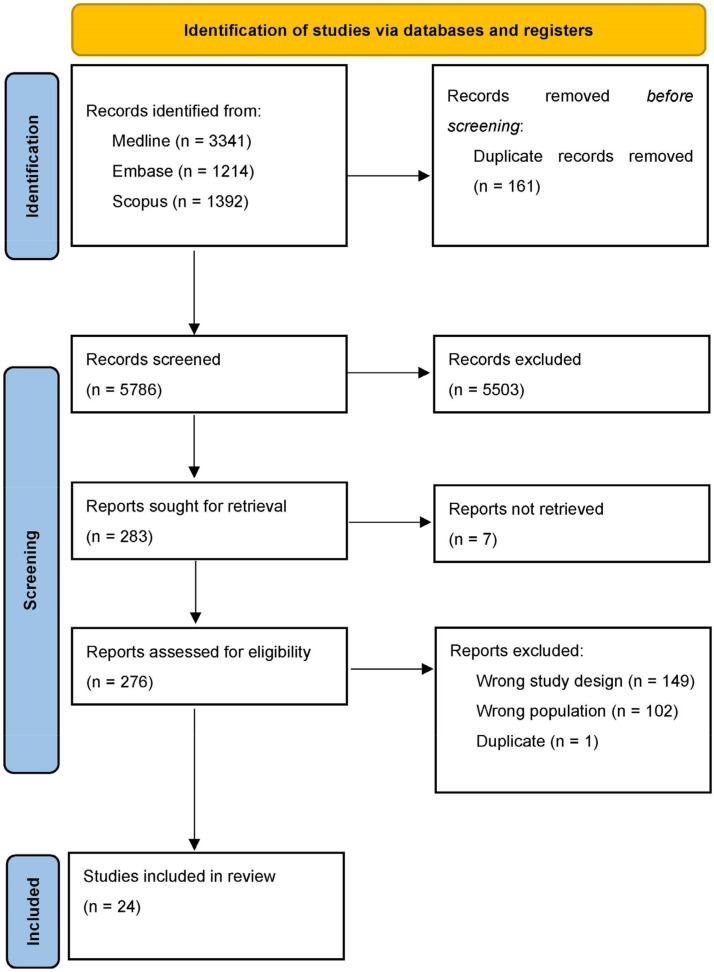
PRISMA diagram of selection process.

**Figure 2 brainsci-13-00609-f002:**
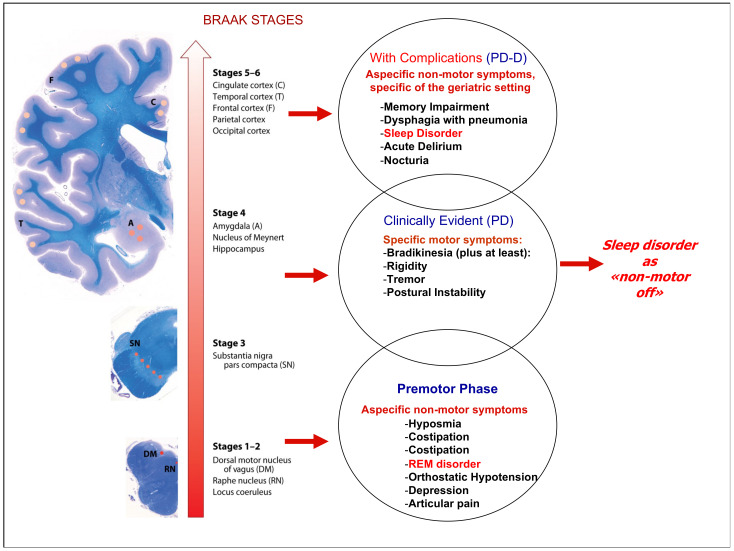
Correlation between the Braak scale in Parkinson’s disease and symptom presentation. In sleep disorders, the indications for pharmacological therapy depend on the multidimensional evaluation of the patient.

**Table 1 brainsci-13-00609-t001:** Summary of the evidence currently available in the literature.

Authors	Parkinson’s Disease Setting	Sleep Assessment	Mean Age (Years)	Design and Methods	Main Conclusion
Moran Gilat et al., 2020 [108]	REM sleep behavior disorder (RBD)	Weekly CIRUS-RBD Questionnaire Video Polisomnography	65	Randomized, double-blind, placebo-controlled, parallel-group trial with an 8-week intervention (melatonin RP 4 mg) and 4-week observation pre- and postintervention	Prolonged-release melatonin 4 mg did not reduce rapid eye movement sleep behavior disorder in PD
Amara et al., 2020 [109]	Subjective sleep quality	Polysomnography. Pittsburgh Sleep Quality Index (PSQI) Epworth Sleepiness Scale (ESS) Psychomotor vigilance task (PVT)	65	Persons with PD were randomized to exercise (supervised 3×/week for 16 weeks) (N = 27) or a sleep hygiene, no-exercise control (in-person discussion and monthly phone calls) (N = 28). Participants underwent polysomnography at baseline and post-intervention. Change in sleep efficiency was the primary outcome, measured from baseline to postintervention	High-intensity exercise rehabilitation improves objective sleep outcomes in PD
Meloni et al., 2021 [110]	REM sleep behavior disorder (RBD)	Video Polisomnography	67	Single-center, randomized, double-blind placebo-controlled crossover trial was performed in a selected population of 18 patients with PD and RBD. The patients received a placebo and 50 mg of 5-HTP daily in a crossover design over a period of 4 weeks	5-HTP is safe and effective in improving sleep stability in PD, contributing to ameliorating patients’ global sleep quality
Hadi et al., 2022 [76]	Subjective sleep quality	Pittsburgh Sleep Quality Index (PSQI) Epworth Sleepiness Scale (ESS) RBD screening questionnaire (RBDSQ)	66	Single-center, double-blind, randomized clinical trial conducted on PD patients with subjective sleep complaints. Eligible patients were randomized 1:1:1 to receive melatonin 3 mg/day, clonazepam 1 mg/day, or trazodone 50 mg/day for 4 weeks. 112 eligible patients were randomized, and 93 participants, melatonin (*n* = 31), trazodone (*n* = 31), and clonazepam (*n* = 31)	Trazodone 50 mg/day, clonazepam 1 mg/day, and melatonin 3 mg/day were all tolerable and effective in improving sleep quality in patients with PD
Peball et al., 2020 [111]	Nonmotor symptoms (NMS)	Epworth Sleepiness Scale (ESS)	65	Placebo-controlled, double-blind, parallel-group, enriched enrollment randomized withdrawal trial; 47 patients with PD with stable motor disease and disturbing NMS underwent open-label nabilone titration (0.25 mg once daily to 1 mg twice daily, phase I). Responders were randomized 1:1 to continue with nabilone or switch to placebo for 4 weeks (phase II)	Improvement of overall NMS burden with nabilone, especially reflected by amelioration of anxiety and sleeping problems
Shin et al., 2019 [74]	REM sleep behavior disorder (RBD)	Korean Epworth Sleepiness Scale (KESS) score 13-item self-reported RBD questionnaire (RBDQ-HK)	66	Four-week, randomized, double-blind, placebo-controlled, parallel group trial in patients with PD and RBD. A total of 40 patients were enrolled, with 20 assigned to receive clonazepam and 20 to receive the placebo	Both clonazepam and placebo tended toward improvement in pRBD symptoms in patients with PD
Stefani et al., 2021 [112]	REM-sleep behavior disorder (RBD)	Video Polisomnography	71	This was a phase 2 multicenter study in Dementia with Lewy Body or Parkinson’s Disease Dementia (PDD) with video polysomnography (vPSG)-confirmed RBD. After a single-blind placebo run-in period, patients meeting eligibility criteria entered a 4-week double-blind treatment period (1:1 ratio with nelotanserin 80 mg/placebo); 8 Patients with PDD were included in the analyses	No difference between nelotanserin and placebo in RBD behaviors
Garcia-Borreguero et al., 2021 [113]	Restless leg syndrome (RLS)	Medical Outcomes Sleep Scale (MOS)	60	A 2-week double-blind, placebo-controlled crossover study assessed the efficacy of dipyridamole (possible up-titration to 300 mg) in untreated patients with idiopathic restless legs syndrome	Dipyridamole has significant therapeutic effects on both sensory and motor symptoms of restless legs syndrome and sleep
Pierantozzi et al., 2016 [114]	Sleep architecture	Polisomnography	63	Randomized, double-blind, placebo-controlled, parallel-group study to determine the efficacy of rotigotine vs. placebo on polysomnography parameters in moderately advanced PD patients	Rotigotine significantly increased sleep efficiency and reduced both wakefulness after sleep onset and sleep latency compared to the placebo
Schrempf et al., 2018 [115]	Sleep parameters	Polisomnography	69	Single-center, double-blind, baseline-controlled investigator-initiated clinical trial of rasagiline 1 mg/day over 8 weeks in PD patients with sleep disturbances	In PD patients with sleep disturbances rasagiline showed beneficial effects on sleep quality as measured by polysomnography
Trenkwalder et al., 2010 [116]	Early-morning motor function and nocturnal sleep disturbance	15-item Parkinson’s Disease Sleep Scale (PDSS-2)	64	Multinational, double-blind, placebo-controlled trial where 287 subjects with Parkinson’s disease (PD) and unsatisfactory early-morning motor symptom control were randomized 2:1 to receive rotigotine 2–16 mg/24 h (190) or placebo (97)	Twenty-four-hour transdermal delivery of rotigotine to PD patients with early-morning motor dysfunction resulted in significant benefits in the control of both motor function and nocturnal sleep disturbances
Silva-Batista et al., 2017 [117]	Sleep quality	Pittsburgh Sleep Quality Index (PSQI)	64	Randomized controlled trial where 22 subjects with moderate PD were randomly as- signed to a nonexercising control group (*n* = 11) or a resistance training group (*n* = 11)	Resistance training improves sleep quality
Larsson et al., 2010 [118]	Sleep disturbances in Parkinson’s disease dementia (PDD)	Stavanger Sleep Questionnaire Epworth Sleepiness Scale (ESS)	76	Randomized controlled trial of 42 patients (20 memantine group, 22 placebo)	Memantine decreases probable REM sleep behaviour disorder in patients with PDD
Di Giacopo et al., 2011 [119]	REM-sleep behavior disorder (RBD)	RBD episodes were monitored by diaries of bed partners	67	Pilot trial	Rivastigmine was well tolerated in most patients, with minor side effects, mainly related to peripheral cholinergic action, and significantly reduced the mean frequency of RBD episodes during the observation time
Büchele et al., 2018 [120]	Excessive Daytime Sleepiness and Sleep Disturbance	Epworth Sleepiness Scale (ESS) Parkinson’s Disease Sleep Scale-2	62	Double-blind, placebo-controlled crossover trial including 12 patients with Parkinson’s disease	Sodium oxybate significantly improved sleepiness and disturbed nighttime sleep both subjectively and objectively
Chaudhuri et al., 2012 [121]	Nocturnal symptoms	Parkinson’s Disease Sleep Scale	66	A 24-week, Phase III, randomized, double-blind, placebo-controlled, multicenter study	Once-daily ropinirole prolonged-release improves nocturnal symptoms in patients with advanced PD not optimally controlled with levodopa
Adler et al., 2004 [122]	Restless leg syndrome (RLS)	RLS rating scale Epworth Sleepiness Scale (ESS)	60	Double-blind, placebo-controlled, crossover study of ropinirole (0.5 to 6.0 mg/day) for restless legs syndrome (RLS)	Ropinirole was effective and well tolerated for treating the symptoms of RLS
Adler et al., 2002 [123]	Subjective Daytime Sleepiness	Epworth Sleepiness Scale (ESS)	65	Single-site, randomized, double-blind, placebo-controlled crossover study of 21 PD patients. They received either a placebo or modafinil 200 mg/day for 3 weeks, followed by a washout week, then the alternate treatment for 3 weeks	Administration of 200 mg/day of modafinil is associated with few side effects and is modestly effective for the treatment of excessive daytime sleepiness in patients with PD
de Almeida et al., 2021 [124]	REM-sleep behavior disorder (RBD)	Video Polisomnography	57	Phase II/III, double-blind, placebo-controlled clinical trial in 33 patients with RBD and PD. Patients were randomized 1:1 to CBD in doses of 75 to 300 mg or matched capsules placebo and were followed up for 14 weeks	Cannabidiol, as an adjunct therapy, showed no reduction in RBD manifestations in PD patients
Plastino et al., 2021 [125]	REM-sleep behavior disorder (RBD)	RBD-screening questionnaire (RBDSQ) REM—sleep behavior disorder questionnaire-Hong Kong (RBDQ-HS) REM Sleep Behavior Disorder Severity scale (RBDSS) Video-Polisomnography	66	Pilot study of 30 patients with PD and RBD was randomized into two groups (15 subjects each), those that received for a period of 3 months safinamide (50 mg/die) in addition (Group A+) or in the absence (Group B) to the usual antiparkinsonian therapy	Safinamide is well tolerated and improves RBD-symptom in parkinsonian
De Cock et al., 2022 [126]	Insomnia	Parkinson’s disease sleep scale (PDSS) Polysomnography	63	Randomised, multicentre, double-blind, placebo-controlled, crossover trial of 46 patients randomly assigned to receive apomorphine or placebo	Subcutaneous nighttime-only apomorphine infusion improved sleep disturbances according to differences on PDSS score, with an overall safety profile
Ahn etl al., 2020 [69]	Poor sleep quality	Pittsburgh Sleep Quality Index (PSQI) Epworth Sleepiness Scale (ESS)	66	Double-blind, placebo-controlled, multicenter trial to evaluate the efficacy and safety of prolonged-release melatonin (PRM) in Parkinson’s disease (PD) patients with poor sleep quality	PRM is an effective and safe treatment option for subjective sleep quality in PD patients and beneficial effects on sleep quality are associated with improved nonmotor symptoms and quality of life in PD patients
Menza et al., 2010 [78]	Insomnia	Polysomnography	56	Six-week, randomized, controlled trial of eszopiclone and placebo in 30 patients with PD and insomnia	Eszopiclone did not increase total sleep time significantly but was superior to placebo in improving the quality of sleep and some measures of sleep maintenance which is the most common sleep difficulty experienced by patients with PD
Wailke et al., 2011 [127]	Microstructure of sleep in Parkinson’s	Polysomnography	61	There were 32 patients with dopamine-responsive, akinetic-rigid PD, not taking neuroleptic medication, or suffering from dementia were randomized into two groups. Both groups had to withhold their usual dopaminergic medication until after noon. At bedtime, one group received 200 mg controlled-release (CR) levodopa/carbidopa, whilst the other group spent the night in the off state	Levodopa/carbidopa CR has no impact on the altered sleep structure in PD

## Data Availability

Not applicable.

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
