# Peer review of "Clinical Evaluation of Sleep Disorders in Parkinson’s Disease"

_brainsci, 2023, doi:10.3390/brainsci13040609_

Round 1

Reviewer 1 Report (New Reviewer)

The sleep disorders in Parkinson's Disease (PD) are a feature which is interpreted as helpful in differential diagnosis with other parkinsonisms and a vital factor preceding the initiation of motor symptoms in PD. The work could be further improved by:

1. Authors could add a initial perspective on the pathogenesis of PD - e.g. genetic, inflammatory etc. Ref.

Platelet-to-lymphocyte ratio and neutrophil-tolymphocyte ratio may reflect differences in PD and MSA-P neuroinflammation patterns. Neurol Neurochir Pol. 2022;56(2):148-155. doi: 10.5603/PJNNS.a2022.0014. Epub 2022 Feb 4. PMID: 35118638.

A Review of Genetic and Gene Therapy for Parkinson's Disease. Cureus. 2023 Feb 5;15(2):e34657. doi: 10.7759/cureus.34657. PMID: 36909056; PMCID: PMC9991874.

2. Authors should acknowledge diseases with possibly similar manifestation regarding sleep disorders - atypical parkinsonisms and Dementia Lewy Bodies

3. The pharmacological point of view should be expanded e.g. the issue of zolpidem and related drugs.

4. It would valuable to present a perspective on whether the issue of sleep disorders should be interpreted as a reason or consequence of neurodegeneration

5. The future perspectives of intervention in sleep disorders in Parkinson's Disease should be evolved.

Author Response

Reviewer 1

The sleep disorders in Parkinson's Disease (PD) are a feature which is interpreted as helpful in differential diagnosis with other parkinsonisms and a vital factor preceding the initiation of motor symptoms in PD. The work could be further improved by:

 Authors could add a initial perspective on the pathogenesis of PD - e.g. genetic, inflammatory etc. Ref. Platelet-to-lymphocyte ratio and neutrophil-tolymphocyte ratio may reflect differences in PD and MSA-P neuroinflammation patterns. Neurol Neurochir Pol. 2022;56(2):148-155. doi: 10.5603/PJNNS.a2022.0014. Epub 2022 Feb 4. PMID: 35118638.

A Review of Genetic and Gene Therapy for Parkinson's Disease. Cureus. 2023 Feb 5;15(2):e34657. doi: 10.7759/cureus.34657. PMID: 36909056; PMCID: PMC9991874.

We are thankful to Reviewer for Her/His suggestions aiming at improving the manuscript. In accordance with your suggestions we have added the references that have been kindly reported to us.

Authors should acknowledge diseases with possibly similar manifestation regarding sleep disorders - atypical parkinsonisms and Dementia Lewy Bodies

We are thankful to Reviewer for Her/His suggestions aiming at improving the manuscript but we felt that adding a paragraph about dementia with Lewy bodies could divert attention from the main focus of the article.

 The pharmacological point of view should be expanded e.g. the issue of zolpidem and related drugs.

We are thankful to Reviewer for Her/His suggestions aiming at improving the manuscript. The paragraph concerning drugs was the most difficult to manage because very often there was the risk of including details in the discussion that could distract the reader from the main focus of the article. Our goal in this case was to make the paper as streamlined as possible while keeping the main focus and message not which drugs to use, but how to choose them based on the stage of the disease.

 It would valuable to present a perspective on whether the issue of sleep disorders should be interpreted as a reason or consequence of neurodegeneration.

 The future perspectives of intervention in sleep disorders in Parkinson's Disease should be evolved.

We thank the Reviewer for Her/His suggestions aiming to improve the manuscript. As you indicated we have added food for thought for possible future perspectives.

Reviewer 2 Report (New Reviewer)

1.     Grammatical English should be revised. There are uncommon structures in the manuscript that could lead to misunderstandings.

2.     IDEAS

a. Could the authors provide a table with the most common sleep disorders in PD and their prevalence?

b. Could the authors provide a table with the scales already used for assessing sleep in PD?

                                               i.     Pitton Rissardo J, Fornari Caprara AL. Parkinson’s disease rating scales: a literature review. Ann Mov Disord 2020;3:3-22

c. A figure about the pathophysiological explanation for developing sleep disorders in PD.

d. A chapter about REM and non-REM sleep disorders.

e. Levodopa possibly causing abnormal sleep architecture.

It is advised to remove references from the conclusion.

Author Response

Reviewer 2

Grammatical English should be revised. There are uncommon structures in the manuscript that could lead to misunderstandings.

We thank the Reviewer for Her/His suggestions. We have extensively modified the language of our manuscript.

IDEAS

Could the authors provide a table with the most common sleep disorders in PD and their prevalence?

Could the authors provide a table with the scales already used for assessing sleep in PD?

Pitton Rissardo J, Fornari Caprara AL. Parkinson’s disease rating scales: a literature review. Ann Mov Disord 2020;3:3-22

We thank the Reviewer for Her/His suggestions aiming to improving the manuscript.  The main focus of our article is the schematization of the different stages of Parkinson's disease and related sleep disorders. Sleep disturbances during the motor phase of the disease can be classified as off state from levodopa therapy. In this sense we felt that adding tables that move away from the main focus could distract the reader's attention.

A figure about the pathophysiological explanation for developing sleep disorders in PD.

We have summarized the main focus of our article in the graphic abstract, for this reason the addition of a figure that focuses on the pathophysiology of sleep disorders could be misleading because the figure would not underline the importance of the modality of presentation of the sleep disturbances in relation to the different stages of the disease.

A chapter about REM and non-REM sleep disorders.

We thank the Reviewer for Her/His suggestions but we felt that adding a chapter on the topic suggested to us could take the reader's attention away from the main focus of the article.

Levodopa possibly causing abnormal sleep architecture.

We thank the Reviewer for Her/His suggestions but, as already highlighted in the article, at the moment the available evidence indicates that levodopa does not cause changes in the architecture of the sleep.

It is advised to remove references from the conclusion.

We thank the Reviewer for Her/His suggestions. As suggested we have removed references from the conclusions.

Round 2

Reviewer 1 Report (New Reviewer)

Authors implemented most of my comments, however I believe it would be valuable to also acknowledge the significance of sleep disturbances in atypical parkinsonisms often overlapping with Parkinson's Disease

Ref. 

Sleep disturbances in progressive supranuclear palsy syndrome (PSPS) and corticobasal syndrome (CBS). Neurol Neurochir Pol. 2023 Mar 17. doi: 10.5603/PJNNS.a2023.0019. Epub ahead of print. PMID: 36928793.

This manuscript is a resubmission of an earlier submission. The following is a list of the peer review reports and author responses from that submission.

Round 1

Reviewer 1 Report

This manuscript entitled “type of sleep disorders in parkinson disease” provides a review of the sleep disturbances found in Parkinson’s disease and summarizes the evidence for various treatments.  The topic that the authors have worked to summarize is a very important one deserving of a comprehensive review, though the execution of this effort is poor with many typographical errors (spelling and capitalization), poor readability, and a general oversimplification of the discussion without discussion of underlying pathophysiology to tell the larger story.  In terms of the therapeutic recommendations, the literature review is frequently incomplete and simplistic in its interpretations.  Overall, the manuscript needs extensive rework before it could serve its purpose of informing readers about sleep disturbances in PD and their management.

Author Response

Dear Reviewer 1,

Thank you for your suggestions.

We now estensively revised the manuscript according to your suggestions

We hope that the paper could be suitable for publication on Brain Sciences

Best Regards,

Reviewer 2 Report

The manuscript reviews type of sleep disorders in Parkinson’s disease. The title suggests that authors will review all the literature available in the context of sleep disorders and parkinson’s disease, however, the article is lacking the objective. In section 2, authors talked about sleep disorders in one huge paragraph, rather they should have divided the section and described and discussed each type in detail. Authors have done fair work to review this topic, but it need polishing to be comprehensive to the audience before it can be published. Additionally, it needs major improvement in language as well.

Below are some my comments-

1.      Capitalize ‘t’ in the title and its Parkinson’s not parkinson.

2.      Syntax and punctuation error can be found throughout the manuscript.

3.      Abstract- what does author mean by quantity of sleep?

4.      Author mentions the article as review article, where as the heading say ‘perpective’, dear editor kindly see into that.

5.      Kindly use better keywords, more appropriate to the context of the article.

6.       Line 46- plethora of drugs are not available for treatment of PD.

7.      Line 56- authors says it’s a systemic review, then they have to provide all the data for information retrieval they used (like keywords used, search engine, years of articles referred to).

Author Response

Dear Reviewer 2,

Thank you for your suggestions.

We now estensively revised the manuscript according to your suggestions

We hope that the paper could be suitable for publication on Brain Sciences

Best Regards,

Round 2

Reviewer 1 Report

With this revision the authors addressed typographical errors and conducted a number of wording edits which in my view have not improved readability, and the authors made no larger content-based edits to address the overly simplistic conclusions drawn throughout the paper.